# Emerging Role of Non-Coding RNAs in Aortic Dissection

**DOI:** 10.3390/biom12101336

**Published:** 2022-09-21

**Authors:** Wei Ding, Ying Liu, Zhe Su, Qi Li, Jianxun Wang, Yufang Gao

**Affiliations:** 1The Affiliated Hospital of Qingdao University, Qingdao Medical College, Qingdao University, Qingdao 266021, China; 2School of Basic Medical Sciences, Qingdao Medical College, Qingdao University, Qingdao 266071, China; 3Institute for Translational Medicine, The Affiliated Hospital of Qingdao University, Qingdao Medical College, Qingdao University, Qingdao 266021, China; 4School of Nursing, Qingdao Medical College, Qingdao University, Qingdao 266071, China

**Keywords:** aortic dissection, micro RNA, long non-coding RNA, circular RNA, biomarker, therapeutic target

## Abstract

Aortic dissection (AD) is a fatal cardiovascular acute disease with high incidence and mortality, and it seriously threatens patients’ lives and health. The pathogenesis of AD mainly includes vascular inflammation, extracellular matrix degradation, and phenotypic conversion as well as apoptosis of vascular smooth muscle cells (VSMCs); however, its detailed mechanisms are still not fully elucidated. Non-coding RNAs (ncRNAs), including microRNAs (miRNAs), long non-coding RNAs (lncRNAs), and circular RNAs (circRNAs), are an emerging class of RNA molecules without protein-coding ability, and they play crucial roles in the progression of many diseases, including AD. A growing number of studies have shown that the dysregulation of ncRNAs contributes to the occurrence and development of AD by modulating the expression of specific target genes or the activity of related proteins. In addition, some ncRNAs exhibit great potential as promising biomarkers and therapeutic targets in AD treatment. In this review, we systematically summarize the recent findings on the underlying mechanism of ncRNA involved in AD regulation and highlight their clinical application as biomarkers and therapeutic targets in AD treatment. The information reviewed here will be of great benefit to the development of ncRNA-based therapeutic strategies for AD patients.

## 1. Introduction

Aortic dissection (AD) is a serious acute cardiovascular disease that seriously threatens patients’ lives and health, bringing a heavy burden on individuals, families, and society [1]. The incidence and mortality rates of AD are increasing yearly. According to the latest statistics from the Global Burden of Disease Project, the total number of years of life lost in 2019 has reached 3.32 million, with nearly 172,000 deaths occurring from the disease [2]. Among them, the mortality rate of acute aortic dissection (AAD) was extremely high, especially Stanford type A acute aortic dissection (AAAD) [3]. In the first 24 h, the mortality of untreated AAAD patients was 1–2% per hour, almost 50% within a week [4]. Due to improvements in surgical techniques, surgical mortality in AAAD patients decreased from 25% to 18%, but the overall mortality was still as high as 57% [5]. Among AD patients, 67% showed type AAD, while the remaining 33% presented type B (male:female = 2:1; mean age: 63 years) [5]. AD has a high misdiagnosis rate, and AD patients receiving surgical treatment present a high incidence of postoperative complications [6]. Studies have shown that 5% to 30% of AD patients receiving surgical treatment and discharged from hospital still die according to a follow-up study conducted several years ago [7]. The regulatory mechanisms of AD development are complex and are still not fully understood. Therefore, further investigations are of great importance for AD patients in the development of individual diagnoses and therapeutic strategies.

Non-coding RNAs (ncRNAs) are a large group of RNA molecules without protein-coding ability that make up more than 90% of all genomic RNA in mammalians [8,9,10]. According to their function, ncRNA is divided into two categories: housekeeping ncRNA (mainly tRNA and rRNA participating in protein synthesis) and regulatory ncRNA (including intronic RNA, microRNA (miRNA), long-chain non-coding RNA (lncRNA), circular RNA (circRNA), and extracellular RNA) [11,12]. Regulatory ncRNAs have been observed in a large number of organisms, and widely participate in complex biological functions by regulating their target gene expression or by affecting the function of proteins and nucleic acids [13]. NcRNAs play crucial roles in a variety of fundamental cellular processes, including growth, differentiation, proliferation, transcription, post-transcriptional modifications, apoptosis, and cellular metabolism [14,15,16]. In general, ncRNA naturally connects genetic networks to influence various basic protein effectors that drive specific cellular biological responses and determine cell fate. Therefore, ncRNA dysregulation contributes to the development of many diseases, including cancers, neurological diseases, growth diseases, and cardiovascular diseases [17,18,19,20]. In recent years, a growing amount of evidence has suggested that ncRNAs are crucial regulators of AD. They play essential roles in the occurrence and development of AD by directly or indirectly regulating the expression of their target genes [21,22]. In addition, some of ncRNAs exhibit great potential as diagnostic and prognostic biomarkers and therapeutic targets in AD [23].

In this review, we focus on the regulatory mechanisms of ncRNAs in AD and highlight the possibility of the clinical application of ncRNAs in AD treatment. This information may provide an opportunity to develop novel ncRNA-based therapeutic interventions for AD patients.

## 2. Overview of AD

### 2.1. Classification of AD

AD is a prototype of acute aortic syndrome, which also includes intramural hematoma, penetrating atherosclerotic aortic ulcer, and thoracic aortic rupture [24]. AD means that the blood flows from where the aortic intima ruptures into the inner membrane, resulting in aorta-outer membrane separation and the formation of the true cavity and false cavity [25]. As the blood continues to flow into the aortic wall, the inner membrane valve extends from the initial tear or bleeding site along the forward and retrograde directions and even to the collateral artery [26]. Severe AD patients can die from aortic rupture. According to the onset time, AD can be divided into acute AD (patients within two weeks of onset), chronic AD (patients with a disease course over three months), and subacute AD (patients with a disease course between two weeks and three months) [27,28]. Based on the site and extent of dissection, there are two common classification methods for AD: DeBakey and Stanford (Figure 1). The DeBakey system distinguishes AD according to their origin and degree: type DeBakey I involves the proximal aorta, aortic arch, and thoracic descending aorta, type DeBakey II involves only the ascending aorta, and type DeBakey III involves the descending thoracic and abdominal aorta. In contrast, the Stanford system divides AD according to the involvement of the ascending aorta: type Stanford A (type DeBakey I and type DeBakey II) involves the ascending aorta, which usually requires rapid open surgical repair; type Stanford B (type DeBakey III) involves the descending aorta and is usually treated by endovascular repair and/or medication [26,29,30] (Figure 1). AD has the most common symptoms of chest pain and back pain [5]. Moreover, controlling hypertension and cigarette smoking, the two major risk factors for AD [31,32], may reduce the incidence of AD.

### 2.2. Pathological Characteristics of AD

The aorta is the largest blood vessel in the human body and can be divided into four anatomical areas: the ascending aorta (starting from the heart and the coronary artery branches from the ascending aorta), the aorta arch (bending above the heart, rotating to the posterior chest wall, producing branches supplying the head, neck, and arms), the descending thoracic aorta (extending through the posterior thorax near the spine), and the abdominal aorta (an aortic region away from the diaphragm; most of the main abdominal branches are further divided into paired iliac arteries in the lower abdomen) [26,33,34]. The normal ascending aorta is 22–36 mm in diameter, while the descending thoracic aorta is 20–30 mm in diameter. The diameter of the abdominal aorta is generally 20 mm. The aorta expands and contracts (also known as the Windkessel effect) with each heartbeat. The aortic root is where the aorta is attached to the heart and includes the aortic valve, with a normal diameter of 30–35 mm. The aortic wall mainly includes the inner layer, middle layer, and outer layer. Vascular smooth muscle cells (VSMCs), elastic fibers, and collagen fibers constitute the middle layer of the aortic wall [35,36,37].

Weak or disrupted structures of the aortic wall, especially the middle part, underlie the pathophysiology of AD. Aortic syndrome-associated genetic disorders are often autosomal dominant and can affect the growth of connective tissue, subsequently causing abnormal aortic structures. Therefore, patients with the aforementioned genetic disorders are susceptible to AD. Marfan syndrome (MFS) is associated with FBN1 mutations [38]; Ehlers–Danlos syndrome (EDS) is relevant to the COL3A1 mutation [39]; Loeysؘ–Dietz syndrome (LDS) is related to TGFBR1 or TGFBR2 mutations as well as SMAD3 or TGFB2 mutations [40]. Aortic valve lobular malformation has an incidence of 1–2% and is also reported to be closely related to AD development mainly due to the lack of elastic fiber components and the increased release of matrix metalloproteinases (MMP) [41]. The most common AD-related non-syndromic mutations are located on the smooth muscle α actin gene (ACTA2). The association between mutations in the gene encoding of the VSMC contractile phenotype and AD suggests that the tension and function of smooth muscle cells play an important role in the aortic wall’s pressure response [42].

### 2.3. Pathogenesis of AD

The pathogenesis of AD mainly includes vascular inflammation, extracellular matrix degradation, phenotypic transformation, and apoptosis of VSMCs and vascular endothelial cells [43,44]. A variety of inflammatory cells (including lymphocytes, macrophages, mast cells, and neutrophils) actively participate in tissue damage, repairment, and remodeling and are detected in the mediators and outer membranes of thoracic AD and abdominal aortic aneurysm tissue [45,46]. In addition, maladjustment of the interaction between fibroblasts with high lysine oxidase (*LOX*) content and smooth muscle cells is one of the causes of AD [47]. Recent studies have shown that several inflammatory factors promote aortic inflammation and aortic aneurysms through different mechanisms. For example, interleukin-1 (IL-1) plays a key role in the formation of thoracic AD by changing the expression of matrix metalloproteinase-2 (MMP-2) and matrix-9 (MMP-9), causing elastic fiber breaks and changing aortic wall stress or strain [48]. It has also been shown that centrogranulocyte-derived MMP-9 triggers acute AD [49].

VSMCs play an important role in maintaining aortic wall homeostasis. Therefore, dysfunctional VSMCs are considered to be a leading factor in AD onset, mainly including apoptosis and phenotypic switching. VSMC phenotypic transformation is characterized by the transformation of a contractile phenotype into a synthetic phenotype. VSMCs mainly exist in the aortic wall with both synthetic and contractile phenotypes with their own characteristics and being complementary. They shift from hyperplastic and synthetic phenotypes in embryonic tissues to quiescent and contractile phenotypes in adult blood vessels. Synthetic VSMCs have the functions of migration, proliferation, and secretion, and produce numerous extracellular matrix proteins, mainly MMP and other cytokines; contractile VSMCs have rich contractile proteins and smooth muscle cells, enabling them to have strong contractility and playing a key role in maintaining vessel wall tension and compliance. Contractile VSMCs are mainly found in the healthy vessel wall and can be converted to synthetic VSMCs when the vascular wall is stimulated by trauma, hypertension impact, and atherosclerosis [50], which is a major pathophysiological alteration in many cardiovascular diseases (including atherosclerosis) [51,52] and high blood pressure [53]. This alteration was found not only in AD and aneurysm disease patients but also in experimental animal models with aortic damage. α-SMA and SM22 α are specific markers of contractile VSMCs [54], while OPN is a specific marker of synthetic VSMCs. Phenotypic transformation of smooth muscle cells plays an important role in the occurrence and development of AD.

Further elucidation of AD pathogenesis and molecular mechanisms will be of great benefit to AD prevention and can provide new biological diagnostic markers and therapeutic targets for AD. Here, we mainly focus on the underlying mechanisms of non-coding RNA (ncRNA) involved in AD regulation, and highlight the great potential of ncRNA as a biological marker and therapeutic target of AD.

## 3. Implications of NcRNAs in AD Progression

With the continuous development of high-throughput sequencing technologies, an increasing number of ncRNAs have been identified in AD. They are widely distributed in the serum, platelets, and tissues of AD patients. NcRNAs play crucial roles in the regulation of AD occurrence and development by targeting specific genes involved in the pathogenesis of AD. Here, we summarize the dysregulated ncRNAs involved in AD progression (Table 1).

### 3.1. MiRNAs and AD

MiRNAs are a large class of short RNA molecules with 19–24 nucleotides in length, modulating the post-transcriptional silencing of the target gene [87,88]. It has been reported that miRNAs can regulate the expression of many genes involved in functional interaction pathways by targeting their mRNAs [89]. In mammals, miRNAs control the activity of about 50% of protein-coding genes and are indispensable regulators in a variety of physiological and pathological processes by modulating cell proliferation, differentiation, apoptosis, and migration [90,91,92]. Therefore, the dysregulation of miRNAs contributes to the occurrence and development of a variety of diseases, including cancer, cardiovascular disease, aneurysm, Kawasaki disease, AD, venous thrombosis, and microvascular complications of diabetes [91,93,94,95,96,97,98,99]. The role of miRNAs in AD has become a popular research topic in recent years. An increasing amount of evidence suggests that miRNAs may play crucial roles in the occurrence and development of AD via distinct mechanisms (Figure 2).

#### 3.1.1. Let-7b, miR-15a /23a, and hcmv-miR-US33-5p

It has been reported that let-7b, miR-15a, miR-23a, and hcmv-miR-US33-5p are significantly upregulated in AD patients. Among these, miR-15a achieves the highest diagnostic accuracy for AD, indicating its great potential as a diagnostic biomarker for AAD and SAD [100].

#### 3.1.2. MiR-21

VSMC phenotypic transformation contributes to aneurysm formation and dissection. Mutations in contraction-related genes, such as Myh11 and ACTA2, can cause aneurysms and dissection [101]. Abnormal transformation growth factor-β (TGF-β) signaling induces a VSMC phenotypic transformation, which is also associated with aneurysm development and dissection in humans and mice. In patients with MFS, VSMCs exhibit a contractile phenotype due to enhanced TGF-β signaling, resulting in vascular stiffness [102]. Huang et al. found that the level of miR-21 associated with phosphorylated extracellular signal-regulated protein kinase (p-ERK) and phosphorylated c-Jun N-terminal kinase (p-JNK) was higher in thoracic aortic aneurysm and dissection (TAAD) lesions. However, Smad3 heterozygous mice showed significant TAAD formation after angiotensin II (Ang II) infusion. The vascular wall was dilated, and the aortic rupture occurred within 23 days of Ang II infusion. Their research found that miR-21 knockdown in Smad3 heterozygous mice increases the Smad7 level and inhibits canonical TGF-β signaling. VSMCs lacking TGF signaling tend to switch from a contractile phenotype to a synthetic phenotype. Silencing of Smad7 with lentivirus prevents TAAD formation in Ang II-induced Smad3 heterozygous mice. This indicates that micRNA-21 knockout results in Ang II-induced signaling abnormalities in TGF-β and promotes thoracic aortic aneurysm and AD formation in Smad3 heterozygous mice [68].

#### 3.1.3. MiR-22

The regulatory mechanism of vascular remodeling has become a research hotspot in AD [103] whose development involves pathologic vascular remodeling and cellular or molecular dysfunction. Xu et al. introduced siRNA plasmids into VSMCs to explore the role of miR-22 and p38 (a mitogen-activated protein kinase) in controlling apoptosis in VSMC cells in vitro. In addition, an AD mouse model was established, and histopathological analysis was performed to assess the regulatory role of miR-22, indicating that miR-22 decreases in the human aorta while VSMC apoptosis increases. The downregulation of miR-22 increases the apoptosis of VSMCs in vitro. Bioinformatic analysis revealed that p38 is a target site of miR-22. Inhibition of p38 expression reverses VSMC apoptosis induced by miR-22 downregulation. Knockdown of miR-22 in AD mouse models significantly promotes AD development. The miR-22′s inhibition of VSMC apoptosis in AD vascular remodeling by targeting p38 suggests that miR-22 may be a novel therapeutic approach for AD regulating apoptosis of VSMCs through the MAPK signaling pathway [104].

#### 3.1.4. MiR-25/26b/29a/155

Su et al. collected serum from 104 AAAD patients and 103 age-matched blood donors. Initial screening was performed using TaqMan low-density arrays, followed by RT-qPCR confirmation, revealing a significant increase in miR-25, miR-29a, and miR-155, but a decrease in miR-26b [105].

#### 3.1.5. MiR-27a

MiR-27a is expressed in many different human diseases, including tumors [106,107,108], atherosclerosis [109], lower limb ischemia [68], and C hepatitis [110]. MiR-27a is involved in pathways critical for endothelial integrity [111,112] and the inhibition of EC inflammation by regulating the nuclear factor-B pathway [113]. Sun et al. found that the miR-27a in the inner membrane of AD samples was lower than that in samples of healthy individuals. The downregulation of miR-27a in endothelial cells is due to the upregulation of the expression of fas-associated protein with death domain (FADD) and the activation of the apoptotic pathway. Because of the enhanced growth differentiation factor 8 (GDF8) in the coculture system supernatant and the inhibition of matrix metalloproteinase 20 (MMP-20), endothelial cells promote VSMC migration following miR-27a downregulation. The AD mouse model shows increased FADD and apoptosis in endothelial cells inducing AD, where miR-27a is stably knocked down by antagomir and thus exerts a protective effect against AD. miR-27a regulates vascular remodeling in AD by targeting endothelial apoptosis and interacting with VSMCs [44].

#### 3.1.6. MiR-30a

*LOX* and its associated gene family members are a group of copper-dependent amine oxidases that cross-link lysyl residues on structural proteins during the formation of proper elastic lamina and collagen fibers [114]. *LOX* plays an important role in the development and function of the cardiovascular system, and *LOX* inactivation is associated with the formation of aortic aneurysm and AD [115]. MiR-30a can act on the targets of different diseases in different tissues. Studies show that targeting Snail-1 by miR-30a may be related to myocardial fibrosis; the average Snail-1 expression level significantly increases in cardiomyocytes and tissues, which can be inhibited by miR-30a [116]. MiR-30 downregulation contributes to endoplasmic reticulum (ER) stress and the associated upregulation of GRP78 in the cardiovascular system. The involvement of miR-30 creates a positive feedback loop in the ER stress signaling pathway [117]. Yang et al. employed human aortic specimens from AD and aneurysms and aortic specimens from cardiac transplanted donors as controls. Rats were pretreated with agomiR-30a or antagomiR-30a, and an empty vector was injected into the control group. *LOX* and elastin expression, as well as the gene expression of miR-30a in VSMCs and human and rat aortic samples, were determined by protein blotting analysis and real-time quantitative polymerase chain reaction. In the AD specimens, miR-30a showed much higher gene expression and a significantly lower protein abundance of *LOX* and elastin (*p* < 0.05). AgomiR-30a transfection significantly reduces luciferase activity in wild-type *LOX* but fails to reduce luciferase activity in the *LOX*-UTR mutant. In cultured VSMCs, agomiR-30a transfection significantly enhances miR-30a gene expression and downregulates the protein abundance of *LOX* and elastin (*p* < 0.05, compared to control groups). In vivo pretreatment with agomiR-30a enhances miR-30a expression and downregulates the protein abundance of *LOX* and elastin in the rat aorta (*p* < 0.05 compared to controls). The stripping rate is significantly high in agomiR-30a-pretreated rats, and the overexpression of miR-30a can promote AD development by reducing the *LOX* content [56].

#### 3.1.7. MiR-107-5p

Wang et al. investigated whether miR-107-5p is highly expressed in acute AD tissues. Moreover, miR-107-5p overexpression promotes VSMC proliferation and inhibits apoptosis. MiR-107-5p inhibits AAD progression by targeting the integral membrane protein 2C (*ITM2C*) [118].

#### 3.1.8. MiR-124

MiR-124 is significantly downregulated in the aortic membrane of AD clinical specimens. It regulates the VSMC phenotype switch by targeting specificity protein 1 (Sp1) expression and plays a key regulatory role in the differentiation, proliferation, and phenotypic switch of human aortic VSMCs [73].

#### 3.1.9. MiR-134-5p

Numerous studies have revealed the significance of miR-134 in the proliferation, differentiation, and migration of cells [119,120,121]. MiR-134 has also been reported to play an inhibitory role in the growth, migration, and tubular formation of vascular endothelial cells in atherosclerosis [122]. Wang et al. selected miRNA microarrays from 12 aortic samples with TAD and 12 controls and identified that miR134-5p is significantly downregulated in TAD tissues. Quantitative PCR detection revealed that miR-134-5p is also significantly reduced in human arteriolar smooth muscle cells. In addition, miR-134-5p is found to inhibit the conversion and migration of the VSMC phenotype reduced by platelet-derived growth factor-BB (PDGF-BB), significantly promoting the expression of VSMC differentiation and contractile markers, such as α-SMA, SM22α, and MYH11. STAT5B and ITGB1 are downstream targets of miR134-5p in human VSMCs and serve as mediators of VSMC phenotypic transformation and TAD progression. Finally, miR-134-5p significantly inhibits aortic dilation and vascular matrix degeneration in TAD mice after angiotensin-induced vascular injury [69].

#### 3.1.10. MiR143/145

In one study, mouse aortic VSMCs were treated in vitro with Ang II of different durations and doses, and the expression levels of miR143/145 and VSMC phenotype marker proteins (i.e., α-SMA and OPN) were determined by quantitative polymerase chain reaction and/or Western blotting. After 12 h of Ang II treatment, the expression of OPN and phospho-p38 significantly increased in VSMCs. Ang II treatment also downregulated both miR143 and miR145 expression. When blocking p38 signaling by pretreatment with SB203580 inhibitors, the expressions of miR143, miR145, and VSMC phenotypic markers were not affected by Ang II. Immunohistochemical staining of aortic tissue from AD patients and healthy donors revealed decreased α-SMA expression in pathological tissue but increased OPN and disordered media smooth muscle cells. Ang II can regulate the expression of the miR143/145 gene cluster and the phenotype switch of VSMCs by the p38 signaling pathway, which may play an important role in AD pathogenesis [123]. Another study showed that miR-145 is involved in AD formation by inducing the proliferation, migration, and apoptosis of VSMCs [60].

#### 3.1.11. MiR-4787-5p and miR-4306

Wang et al. identified the differential expressions of miRNA in the plasma of AAD patients and age-matched healthy volunteers. Nine upregulated and twelve downregulated minute ribonucleic acids were found in the circulating plasma of AAD patients. Quantitative reverse transcription-polymerase chain reaction confirmed the statistical agreement of the expression of two selected minute ribonucleic acids. ROC analysis indicated that miR-4787-5p and miR-4306 were specific and sensitive to the early diagnosis of AAD. Bioinformatics predictions and double-luciferase assays revealed that polycystin-1 (PKD1) and transformed growth factor-1 (TGF-1) are direct targets of miR-4787-5p and miR-4306, respectively. In addition, the protein expression of PKD1 and TGF-1 downstream targets was significantly reduced after the 700 overexpression of miR-4787-5p and miR-4306. These results suggest that miR-4787-5p and miR-4306 serve as potential diagnostic biomarkers of AAD and may participate in AAD pathogenesis [67].

Taken together, these findings strongly support the idea that miRNAs exert their functions in AD progression by modulating the expression of specific target genes. Moreover, some of them also exhibit great potential as biomarkers for AD diagnosis. Further studies are required to identify key regulators by analyzing differentially expressed miRNAs in AD tissues or cell lines, which may provide new insights into miRNA-based therapeutics strategies in AD.

### 3.2. LncRNAs and AD

LncRNA is defined as non-protein-coding transcripts greater than 200 nucleotides [124]. It is transcribed by the action of RNA polymerase II. The 5′ end methylation cap and the 3′ end polymeric A tail are spliced to form a fully mature linear lncRNA [125]. LncRNA can be roughly classified into the following four classes based on their distinct genomic origins [126]: (1) synonymous lncRNA overlapping protein-coding genes by sharing the same promoters; (2) antisense lncRNA located in the antisense direction of protein-coding genes (opposite chains); (3) introns produced by introns of protein-coding genes; and (4) intergenic lncRNA (also known as interchain ncRNA) located between two protein-coding genes. Because of its long sequence, as well as its complex three-dimensional structure and domain, lncRNA has various sites interacting with different macromolecules, including DNA, RNA, and proteins. The range of action involves the epigenetic level, transcriptional level, and post-transcriptional level of genes, and can also play a “sponge” role as a competitive RNA [127]. LncRNA is dynamically expressed in different cell types, diseases, or developmental stages of organisms and plays a regulatory role at almost every step of gene expression and translation [128]. LncRNAs have been shown to be involved in the regulation of AD progression (Figure 2).

#### 3.2.1. XIST

The XIST gene plays an important role in the pathogenesis of type A TAD and restores the new mechanism by which XIST regulates type A TAD by inhibiting miR-17 and regulating the subsequent downstream genes of PTEN, providing new ways for type A TAD treatment. XIST is upregulated in aortic wall tissue and is associated with a prognosis in type A TAD patients. Silencing XIST promotes VSMC proliferation and inhibits VSMC apoptosis, whereas restored XIST shows the opposite effect. Furthermore, mechanistic studies have shown that XIST contains the binding site of miR-17, whose downregulation reverses the increase in cell proliferation and apoptosis attenuation induced by XIST silencing. Further studies showed that XIST positively regulates PTEN expression through its competing target, miR-17. Taken together, the knockdown of XIST genes may slow type A TAD progression through miR-17 activation and subsequent downstream PTEN regulation, providing a novel therapeutic target for type A TAD treatment [79].

#### 3.2.2. XIST/ENSG00000269936/lncRNA1421/ENSG00000248508/ENSG00000226530/EG00000259719

Sun et al. examined quantitative real-time polymerase chain reaction (qRT-PCR) using high-throughput sequencing to verify the results. A total of 269 lncRNAs (159 upregulated and 110 downregulated) and 2255 mRNAs (1,294 upregulated and 961 downregulated) were abnormally expressed in human TAD. Among these, ENSG00000269936 upregulation was positively correlated with its adjacent mRNA MAP2K6, while lncRNA-1421 downregulation was positively relevant to mRNA FBLN5TIMP3. The lncRNA-miRNA-mRNA network suggests that the upregulated lncRNA XIST and p21 have similar sequences targeted by has-miR-17-5p. Upregulation of ENSG00000248508, ENSG00000226530, and EG00000259719 was associated with RUNX1 [129].

#### 3.2.3. Lnc-C2orf63-4-1

Zhou et al. indicated that overexpression of Lnc-C2orf63-4-1 significantly reduced Ang II-induced apoptosis, phenotypic switch, and extracellular matrix degradation in vivo and vitro. Transcriptional activator 3 (STAT3) is the primary downstream effector. A dual luciferase assay and RNA antisense purification (RAP) assay showed that Lnc-C2orf63-4-1 reduces STAT3 expression, and STAT3 upregulation reversed the protective effect of Lnc-C2orf63-4-1 on Ang II-mediated vascular remodeling. Lnc-c2orf63-4-1 negatively regulates STAT3 expression and prevents AD. In conclusion, lnc-C2orf63-4-1 plays an important role in vascular homeostasis, and its dysfunction exacerbates Ang II-induced pathological vascular remodeling [76].

#### 3.2.4. LncRNA OIP5-AS1

Wang et al. found that lncRNA OIP5-AS1, as a competitive endogenous RNA (ceRNA) of miR-143-3p, inhibits proliferation and migration and promotes apoptosis of human aortic endothelial cells (HAECs) and human aortic smooth muscle cells (HASMCs); simultaneously, it causes an imbalance of MMP-2/9 and TIMP-2/1 in HASMCs. This promotes the secretion of IL-6, IL-1β, and IL-17A in human aortic adventitial fibroblasts (HAAFs). LncRNA OIP5-AS1 upregulates TUB through sponge miR-143-3p, thereby exacerbating damage to the intima, media, and adventum of the aorta and thereby promoting AD [20].

#### 3.2.5. Linc01278

Linc01278 and ACTG2 expressions are downregulated, and miR-500b-5p expression is upregulated in AD tissues. Linc01278 directly targets ACTG2 via sponge miR-500b-5p. Inhibition of Linc01278 promotes proliferation, migration, and phenotypic transformation of VSMCs. The linc01278-miR-500b-5p-ACTG2 axis has a potential role in the development of diagnostic markers and therapeutic targets for AD [80].

Collectively, these findings indicate that lncRNAs play crucial roles in regulating the progression of AD. However, the exact roles are still inconclusive. Further investigations are still required to elucidate the detailed mechanisms of lncRNA in AD progression.

### 3.3. CircRNAs and AD

CircRNA is a novel class of ncRNA molecules generated from the anti-splicing events of one or two exons, whose 5′ and 3′ ends are covalently linked to form a closed circular structure [130,131,132]. Due to its circular structure, circRNA is not easily degraded by exonucleases and shows greater stability than linear RNA [133,134]. According to its source [135,136], circRNA can be divided into ecircRNA (formed by exon), ciRNA (formed by intron), and EIciRNA (jointly formed by exons and introns). CircRNA has the following features: structurally stable and refractory to degradation by nuclease, tissue- and cell-specific expression and conservative evolution, mostly formed by exons, some contain miRNA binding sites, and most are in the cytoplasm [137]. Functional research of circRNA has focused on the following aspects: serving as “sponge” in counteracting miRNA inhibition of its target gene expression, binding to protein to inhibit its function, and some ciRNA and EIciRNA can bind to RNA polymerase Pol II regulation to regulate maternal gene expression [138]. Studies from this year show that circRNA has close ties to diabetes, neurological diseases, multiple cardiovascular diseases, etc [139]. Growing evidence has revealed that circRNAs are crucial regulators during AD progression (Figure 2).

#### 3.3.1. CircMARK3

Tian et al. explored the differential expression of circRNA and miRNA in human AAAD tissues (*n* = 10) and normal aortic tissues (*n* = 10) using high-throughput RNA sequencing (*n* = 10). RNA-Seq results showed that 506 circRNAs are significantly dysregulated, and subsequent analysis indicated that the tyrosine protein kinase Fgr may play an important role in the occurrence and development of AAAD. Another regulatory molecule upstream of Fgr is found to be circMARK3, indicating that circRNA3 is a potential biomarker for AAAD diagnosis [86].

#### 3.3.2. CircRNA-101238

A circRNA microarray was analyzed in human type A TAD patients and age-matched normal donors. A total of 8173 circRNA genes were tested, with 156 upregulated and 106 downregulated. Quantitative real-time PCR showed increased expression of upregulated hsa-circRNA-101238, hsa-circRNA-104634, hsa-circRNA-002271, hsa-circRNA-102771, hsa-circRNA-104349, COL1A1, and COL6A3, as well as decreased downregulation of hsa-circRNA-102683 and hsa-circRNA-005525. Moreover, hsa-circRNA-101238 may inhibit hsa-miR-320a expression in TAD and increase MMP9 expression [84].

#### 3.3.3. Circ-TGFBR2

Circ-TGFBR2 has been found to be downregulated in AD tissue, whose inhibition promotes proliferation, migration and phenotypic transformation of VSMCs. Circ-TGFBR2 acts as a sponge for miR-29A targeting KLF4. In vivo experiments show that circ-TGFBR2 overexpression can inhibit AD progression, increase the expression of contraction markers, and inhibit the expression of synthetic markers [85].

Taken together, these findings suggest that circRNAs are crucial regulators of AD progression. An increasing amount of differentially expressed circRNAs have been identified in AD, indicating their great potential as diagnostic and prognostic biomarkers for AD patients. These dysregulated circRNAs may contribute to AD progression by sponging miRNAs. However, investigations of their role in AD are still in infancy. Further studies should focus on the underlying mechanisms of circRNAs in AD regulation and their feasibility as biomarkers for AD diagnosis, which may provide new insights for the development of novel therapeutic strategies for AD patients.

## 4. Non-Coding RNAs as Diagnostic Markers in AD

Currently, there is a lack of effective and fast methods for diagnosing AD in clinics. The overlap of AD symptoms with other cardiovascular diseases makes its early diagnosis more difficult. Multilayered helical CT, transthoracic or transesophageal echocardiography, and magnetic resonance angiography have been used to improve the diagnostic rate of AD. However, in most cases, these expensive and complex tests cannot be performed in the emergency room, and the examination is delayed. These issues seriously plague AD in a timely and correct diagnosis. Some protein biomarkers, such as D-dimers, calcium-binding proteins, elastin, CD40L, MPO, MMP-1, and TIMP-1, have been investigated as potential candidate biomarkers for AAD [140,141]. However, the unsatisfactory sensitivity and specificity of these biomarkers limit their further application in routine clinical practice. Therefore, the screening and identification of novel, valuable molecular biomarkers with high specificity and sensitivity are in urgent need for AD diagnosis. In recent years, numerous studies have shown the crucial role of ncRNAs in AD pathogenesis, and their differential expression patterns endow ncRNAs with great potential as promising biomarkers for the diagnosis of AD patients (Table 2). For instance, Su et al. found that miR-25, miR-29a, and miR-155 were markedly upregulated in serum samples from AAAD patients, whereas miR-26b was significantly downregulated. In the discovery cohorts, the 4-miRNA panel showed high accuracy in predicting those who were likelier to develop AAAD [105], suggesting the potential of the 4-miRNA panel as a biomarker for AAAD diagnosis. Dong et al. showed that let-7b, miR-15a, miR-23a, and hcmv-miR-US33-5p were highly expressed in AAD patients. Among these, miR-23a exhibited the highest diagnostic accuracy for AAD [100], indicating its potential as a diagnostic biomarker for AAD. In another study, Wang et al. revealed that the AUC values for miR-4787-5p and miR-4306 were 0.898 and 0.874, respectively, whereas the combination of the two miRNAs had an AUC of 0.961, indicating that the combination of miR-4787-5p and miR-4306 had better specificity and sensitivity for the early diagnosis of AAD [67]. Moreover, Tian et al. demonstrated that the level of circMARK3 in serum distinguished AAAD patients from the normal group, with an AUC of 0.9344, 90.0% sensitivity and 86.7% specificity [86], indicating that circMARK3 might be a promising biomarker for identifying AAAD patients. These findings strongly suggest that ncRNAs have great potential as diagnostic biomarkers for AD. However, the sample size of relevant studies is still small, and high-quality samples and larger patient cohorts are required to further confirm their clinical application in AD treatment.

## 5. Conclusions and Perspectives

NcRNAs, including miRNAs, lncRNAs, and circRNAs, play crucial roles in AD pathological processes by modulating the expression of target genes involved in VSMC phenotypic transformation, pathological vascular remodeling, and extracellular matrix degradation. With the deepening of investigations into the role of ncRNAs in AD, their underlying mechanisms are being clarified. These findings strongly support that ncRNAs possess great potential clinical value for the reduction of AD risk and early diagnosis and treatment of this disease. At present, relevant studies on AD regulation by ncRNA mainly focus on miRNAs, lncRNAs, and circRNA, and especially on miRNAs. In the future, the molecular regulation mechanism of AD by lncRNAs and circRNAs should be further explored, and that by other ncRNAs is expected, such as repeat-associated short interfering RNAs and Piwi-interacting RNA. In addition, the current research on AD pathogenesis mainly focuses on the impact on VSMCs. With the advancement of molecular biology technology, some studies have comprehensively evaluated aortic wall cell composition by applying single-cell RNA sequencing technology [146]. A further deep investigation is embraced into other components of the aortic wall, such as endothelial cells, fibroblasts, and macrophages, through that technology. Furthermore, the pathological mechanism of ncRNAs participating in the occurrence and development of AD in different cells was discussed. With the fast development of high-throughput sequencing technologies, an increasing number of ncRNAs in AD have been identified, and their important functions will be revealed. Therefore, we believe that ncRNAs will be widely used in the early diagnosis, prognosis, and clinical treatment of AD in the future.

## Figures and Tables

**Figure 1 biomolecules-12-01336-f001:**
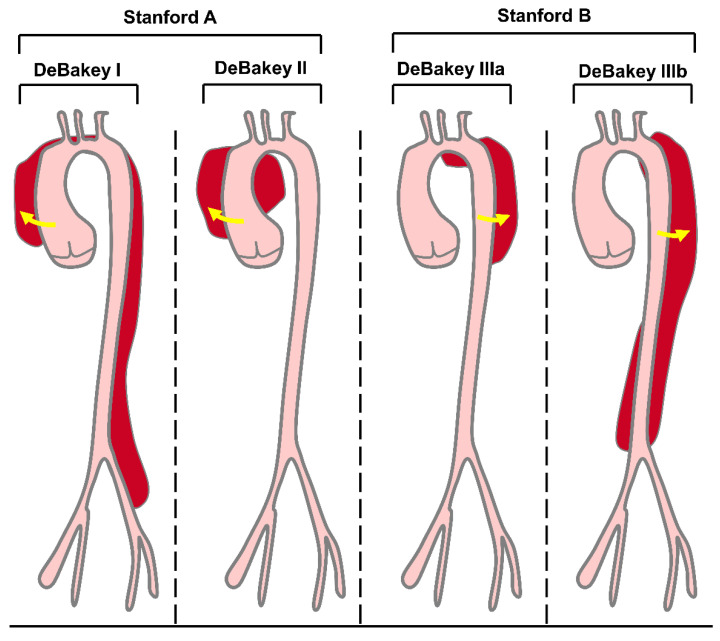
DeBakey and Stanford classifications of AD. The DeBakey classification is based on origin and degree. Type DeBakey I dissections are the most extensive, involving the proximal aorta, aortic arch, and thoracic descending aorta. DeBakey II dissections are the least extensive and involve only the ascending aorta. DeBakey III dissections spare the ascending aorta. Stanford classification is based entirely on the involvement of the ascending aorta. Stanford A dissections involve the ascending aorta, whereas Stanford B dissections involve the descending aorta.

**Figure 2 biomolecules-12-01336-f002:**
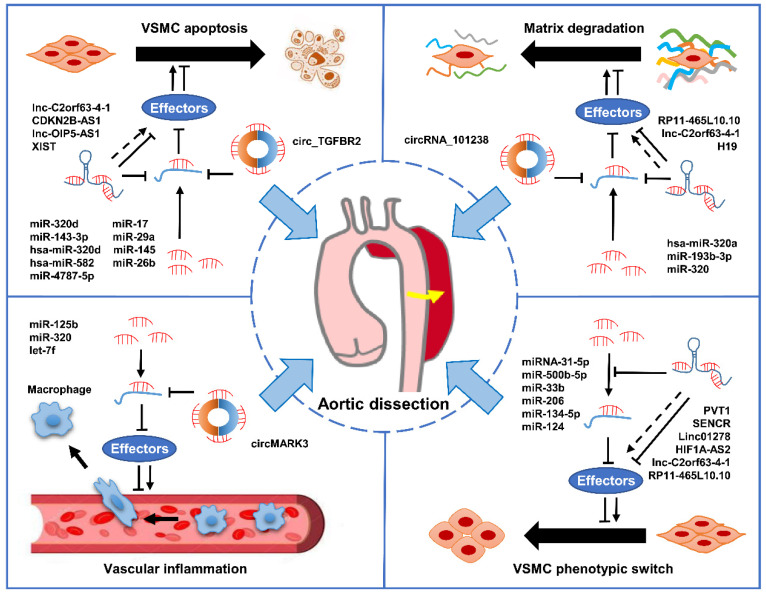
The mechanisms of ncRNAs involved in AD progression. The dysregulation of miRNAs, lncRNAs, and circRNAs contributes to AD progression by modulating crucial biological progresses, including phenotypic transformation and apoptosis of VSMCs, extracellular matrix degradation, and vascular inflammation.

**Table 1 biomolecules-12-01336-t001:** Role of ncRNAs in AD progression.

ncRNAs	Key Messages	References
miRNA-31-5p	MiR-31-5p overexpression significantly inhibited aggravated pathological VSMC phenotypical switch and AAD by downregulating myocardin.	[55]
miR-30a	MiR-30a overexpression facilitated the development of AD by targeting *LOX*.	[56]
hsa-miR-320d, hsa-miR-582	The overexpression of hsa-miR-320d and hsa-miR-582 significantly enhanced staurosporine and TNFα-induced apoptosis of VSMCs by directly targeting the 3′ UTR region of *TRIAP1* and *NET1*.	[57]
miR-145	The overexpression of miR-145 facilitated VSMC proliferation and inhibited cell apoptosis by decreasing CTGF expression.	[58]
	MiR-145 suppressed phenotypic transformation of VSMCs to prevent AD by downregulating KLF4.	[59]
	MiR-145 overexpression promoted the proliferation, migration, and apoptosis of VSMCs by targeting SMAD3.	[60]
miR-21-5p	MiR-21-5p might be involved in ATAAD progression by targeting JAK2, PDGFA, TGFB1, VEGFA, and TIMP3, TIMP4, and SERPINE1.	[61]
miR-29, miR-30	MiR-29 and miR-30 families might play a crucial role in TAD by targeting the focal adhesion and the MAPK signaling pathways.	[62]
miR-133a	The upregulation of miR-133a by APN inhibits pyroptosis pathway in VSMCs via targeting NLRP3, which potentially rescues AAD.	[63]
miR-26b	The overexpression of miR-26b promoted the proliferation of VSMCs and inhibited cell apoptosis by targeting HMGA2 and TGF-β/Smad3 signaling pathway, thereby contributing to the development of TAAD.	[64]
miR-140-5p	MiR-140-5p overexpression suppressed the proliferation, migration, and invasion of VSMCs by directly targeting NCKAP1.	[65]
miR-320	MiR-320 was involved in AD development by regulating the expression of MMPs at the post-transcriptional level via macrophages.	[66]
miR-4787-5p, miR-4306	MiR-4787-5p and miR-4306 might play a role in the pathogenesis of AAD by targeting PKD1 and TGF-β1, respectively.	[67]
miR-21	MiR-21 might be involved in the regulation of TAAD progression by targeting SMAD7 and canonical TGF-β signaling pathway.	[68]
miR-134-5p	The overexpression of miR-134-5p significantly facilitated VSMC differentiation and suppressed PDGF-BB-induced VSMC phenotypic switch and migration by targeting STAT5B and ITGB1.	[69]
miR-146a-5p	MiR-146a-5p promoted VSMC proliferation and migration by targeting SMAD4, thereby contributing to the development of AD.	[70]
miR-27a	MiR-27a was involved in vascular remodeling by inhibiting endothelial cell apoptosis via targeting FADD and HASMC migration via targeting GDF8 and MMP20 in AD.	[44]
miR-4787-5p	MiR-4787-5p overexpression significantly induced VSMC apoptosis by directly targeting PKD1 and suppressing the PI3K/Akt/FKHR pathway, resulting in the development of AD.	[71]
miR-193a-3p	The overexpression of miR-193a-3p promoted VSMC proliferation and migration by targeting ACTG2.	[72]
miR-124	MiR-124 overexpression inhibited the proliferation and phenotype switch of VSMCs by targeting the 3′-untranslated region of sp1.	[73]
miR-125b	The upregulation of miR-125b induced by ATP7A dysfunction increased proinflammatory cytokine expression, aortic macrophage recruitment, MMP-2/9 activity, elastin fragmentation, and vascular smooth muscle cell loss by targeting Suv39h1 and TNFAIP3.	[74]
lncRNA H19 miR-193b-3p	LncRNA H19 promoted the proliferation and migration rate of HASMCs by sponging miR-193b-3p, thereby contributing the development of AD.	[75]
lnc-C2orf63-4-1	The overexpression of lnc-C2orf63-4-1 significantly attenuated Ang II-induced apoptosis, phenotypic switching of VSMCs and degradation of extracellular matrix by directly decreasing STAT3 expression.	[76]
CDKN2B-AS1 miR-320d	CDKN2B-AS1 overexpression significantly facilitated the apoptosis and inhibited the proliferation of VSMCs by upregulating STAT3 via sponging miR-320d during TAD development.	[77]
SENCR	The overexpression of SENCR inhibited VMSC proliferation and migration and suppressed aortic dilatation by upregulating myocardin via sponging miR-206.	[78]
XIST miR-17	XIST overexpression inhibited VSMC proliferation and induced VSMC apoptosis by upregulating PTEN via sponging miR-17, thereby contributing to the progression of TAAD.	[79]
linc01278 miR-500b-5p	The overexpression of linc01278 inhibited VSMC proliferation and phenotypic switching by upregulating ACTG2 via sponging miR-500b-5p during AD development.	[80]
lnc-OIP5-AS1 miR-143-3p	Lnc-OIP5-AS1 inhibited the proliferation and mobility, but facilitated apoptosis of HAECs and HASMCs via sponging miR-143-3p in the development of AD.	[81]
HIF1A-AS2 miR-33b	The overexpression of HIF1A-AS2 promoted the proliferation and migration, while promoting the phenotypic switching of SMCs by upregulating HMGA2 via sponging miR-33b.	[22]
PVT1	PVT1 knockdown inhibited the proliferation, migration, and phenotypic switch of HASMCs by targeting miR-27b-3p.	[82]
RP11-465L10.10	RP11-465L10.10 overexpression promoted VSMC phenotype switching and MMP9 expression via the NF-κB signal pathway.	[83]
hsa_circRNA_101238 hsa-miR-320a	Hsa_circRNA_101238 might play a role in TAD by upregulating MMP9 via targeting hsa-miR-320a.	[84]
circ_TGFBR2 miR-29a	Circ_TGFBR2 overexpression suppressed the proliferation and migration of AD-VSMCs by upregulating KLF4 via sponging miR-29a, leading to the inhibition of AD progression.	[85]
circMARK3	CircMARK3 might be involved in the occurrence and development of AAAD by targeting tyrosine-protein kinase Fgr.	[86]

**Table 2 biomolecules-12-01336-t002:** NcRNAs as diagnostic biomarkers in AD.

NcRNAs	Alteration	Potential Values	References
miR-4787-5p and miR-4306	Up	MiR-4787-5p and miR-4306 were specific and sensitive biomarkers for the early diagnosis of AAD. The AUC values for miR-4787-5p and miR-4306 were 0.898 and 0.874, respectively. The AUC for the combination value of miR-4787-5p and miR-4306 was 0.961.	[67]
miR-25 miR-29a miR-155 miR-26b	Up Down	The distinct profile of 4-miRNA can act as a noninvasive biomarker for AAAD diagnosis, with an AUC of 0.995, 96.00% sensitivity, and 100.00% specificity.	[105]
miR-486	Up	MiR-486 may serve as a non-invasive biomarker of aortic wall degeneration, and its dysregulation was associated with high risk of dissection and rupture in patients with bicuspid aortic valve.	[142]
miR-193a-3p	Up	MiR-193a-3p/ACTG2 axis was involved in AD pathogenesis by regulating phenotypic switching of VSMCs and may act as a promising diagnostic biomarker of AD.	[72]
has-miR-4313 has-miR-933 has-miR-1281 has-miR-1238	Up	The fold change of the four miRNAs was striking, ranging from over 10- to 40-fold increase in plasma compared with healthy control subjects, suggesting their potential as diagnostic biomarker for discrimination between AAD cases and disease-free cases.	[143]
hsa-miR-320d hsa-miR-582	Down	MiR-320d and miR-582 were decreased by 72% (*p* < 0.0005) and 51% (*p* < 0.05), respectively, in AD patients compared to normal control.	[57]
hsa-miR-636	Up	Hsa-miR-636 was significantly upregulated in the AAD versus control comparison (3.3-fold, *p* = 0.012)	[144]
miR-15a miR-23a	Up Up	MiR-15a was a promising diagnostic biomarker for AAD, with an AUC of 0.761, 75.7% sensitivity, and 82.5% specificity. MiR-23a was a promising diagnostic biomarker for AAD, with an AUC of 0.925, 91.9% sensitivity, and 85.7% specificity.	[100]
miR-107-5p	Up	MiR-107- 5p may be a diagnostic biomarker for AD by inhibiting the progression of acute AD via targeting *ITM2C*.	[118]
miR-15a miR-23a let-7b US33-5p	Up	Four miRNAs, including miR-15a, miR-23a, let-7b, and US33-5p, were significantly upregulated in the AAD. The four miRNAs showed sensitivity of 75.7%, 91.9%, 79.4%, and 73.5%, respectively. The specificity values were 100%, 85.7%, 92.9%, and 85.7%, respectively. The corresponding AUCs were 0.855, 0.925, 0.887, and 0.815, respectively.	[145]
circMARK3	Up	CircMARK3 may serve as an effective diagnostic biomarker for AAAD, with an AUC of 0.9344, 90.0% sensitivity, and 886.7% specificity.	[86]

## Data Availability

Not applicable.

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
