# Peer review of "Emerging Role of Non-Coding RNAs in Aortic Dissection"

_biomolecules, 2022, doi:10.3390/biom12101336_

Round 1

Reviewer 1 Report

Ding et al. summarized the emerging role of non-coding RNAs in aortic dissection in this Review article. Manuscripts covered most of the aspects of ncRNAs in aortic dissection. They described the recent advance in this field and summarized the story with future directions. This manuscript is impressed. However, several issues should be further clarified.

Comments:

1. Please check the whole text to correct grammar errors. The English writing needs to be improved. Page1, line 10, two “Correspondence” existed, and the first one should be deleted. Page 1, line 35, “with nearly 10.0 172,000 deaths……” should be corrected.

2. Some statements do not have reference. For instance, the statements in page 2, line 63-67, and the statements in page 11, line 356-357. Relevant works should be added.

3. The conclusion of “circRNAs and AD” is too brief and not clear to the readers. The authors should provide several conclusive points.

Reviewer 2 Report

Manuscript of Emerging role of non-coding RNAs in aortic dissection is very interesting. It's great information as well reviewed in this manuscript could greatly benefit the development of ncRNA-based therapeutic strategies patients with AD. my recommendation is moderate small English changes required.
